# A General Framework for Auditing Differentially Private Machine Learning

**Fred Lu**
Booz Allen Hamilton

**Joseph Munoz**
Booz Allen Hamilton

**Maya Fuchs**
Booz Allen Hamilton

**Tyler LeBlond**
Booz Allen Hamilton

**Elliott Zaresky-Williams**
Booz Allen Hamilton

**Edward Raff**
Booz Allen Hamilton

**Francis Ferraro**
University of Maryland, Baltimore County

**Brian Testa**[*]
Air Force Research Laboratory

## Abstract

We present a framework to statistically audit the privacy guarantee conferred by a differentially private machine learner in practice. While previous works have taken steps toward evaluating privacy loss through poisoning attacks or membership inference, they have been tailored to specific models or have demonstrated low statistical power. Our work develops a general methodology to empirically evaluate the privacy of differentially private machine learning implementations, combining improved privacy search and verification methods with a toolkit of influence-based poisoning attacks. We demonstrate significantly improved auditing power over previous approaches on a variety of models including logistic regression, Naive Bayes, and random forest. Our method can be used to detect privacy violations due to implementation errors or misuse. When violations are not present, it can aid in understanding the amount of information that can be leaked from a given dataset, algorithm, and privacy specification.

## 1 Introduction

Machine learning (ML) is increasingly deployed in contexts where the privacy of the data used to build the model is of concern. ML models are capable of ingesting enormous quantities of data in order to determine meaningful predictive patterns. When such models are built on potentially sensitive inputs such as healthcare [1] or financial data [2], it becomes important to determine to what extent information from the training dataset can be inferred from the model. Such information can be of personal interest when the identity of a data contributor may be compromised based on involvement in the data collection process [3, 4], but it can also lead to regulatory concern about whether a model trained on private data should itself be considered private [5].

Differentially private (DP) machine learning is a theoretically rigorous approach that provides a worst-case privacy guarantee for ML algorithms [6]. Specifically, it provides a mathematical guarantee that the distribution of possible output models based on the input dataset is not significantly altered whether an individual data point is included in the training set or not. This is accomplished by randomizing the output model in a specific manner so that the distributions of the model trained

---

[*]Approved for Public Release; Distribution Unlimited. PA #: AFRL-2022-3247

36th Conference on Neural Information Processing Systems (NeurIPS 2022).

with or without any individual data point are statistically indistinguishable up to a level specified by parameters $\varepsilon, \delta$ that must be set appropriately. While DP mathematically certifies the privacy of a mechanism, the noise added is generally tailored to the worst case scenario.

Previous results attempting to assess the privacy leakage of such models, using attacks ranging from membership inference to model inversion, have inferred that the actual detectable risk of a model procedure is lower than the theoretical bound [7, 8]. While a recent study on DP neural networks indicates that the privacy guarantee of the DP-SGD algorithm is tight under the strongest adversary [9], the privacy risk under realistic datasets and attackers without complete access to the training process still lies below the worst case bound. Across other models, the question remains of whether the privacy violation of a mechanism is lower than specified or whether the attacks previously used are not strong enough. Given that there are no guidelines on how to set $\varepsilon, \delta$ in practice, it would be useful for practitioners to estimate the actual risk they incur in practice and set the parameters accordingly. This is critical when DP for ML is still a nascent field, with limited implementations that can be hard to verify.

Recent works have shown that simple DP mechanisms can be audited effectively [10, 11]. These mechanisms generally map an input vector to a perturbed output vector, and are often used as building blocks of more complicated DP algorithms such as machine learning models. Examples include the Laplace and exponential mechanisms which are respectively used in DP Naive Bayes and random forest [12, 13]. The auditing procedure involves searching for optimal neighboring input sets $a, a'$ and sampling the DP outputs $\mathcal{M}(a), \mathcal{M}(a')$, to get a Monte Carlo estimate of $\varepsilon$. To extend these techniques to audit a machine learning model, the vector inputs $a, a'$ are replaced with datasets $D, D'$ which differ by a single entry. Working in this realm raises important challenges. First, previously effective search methods for neighboring inputs involving enumeration or symbolic search are impossible over large datasets, making it difficult to find optimal dataset pairs. In addition, Monte Carlo estimation requires costly model retraining thousands of times to bound $\varepsilon$ with high confidence.

We propose solutions to these problems to accommodate the general auditing approach to machine learning algorithms. For the first concern, we develop *a set of data poisoning attacks which perturb a single point of dataset $D$*, approximating a worst-case neighboring dataset $D'$ specific to $\mathcal{M}$. To address the second, we present a statistical framework combining elements of Bichsel et al. [11] and Jagielski et al. [14] *to estimate $\varepsilon$ for general ML procedures for smaller sample sizes*. Our techniques involve *improved estimation bounds* and, more significantly, *learning probabilities over general model spaces*, while previous works can only estimate probabilities over a single output (the predictive probability).

These novel contributions form our framework *ML-Audit* for auditing the DP of arbitrary machine learning models. Compared to prior works we can obtain orders of magnitude improvement in the estimation of $\varepsilon$ that are effective across a larger range of $\varepsilon$, more datasets, and more model types. Our framework is compatible with prior model-specific approaches (small neural networks), obtaining similar or improved efficacy. Further still, we provide case studies where our framework detects potential violations in existing implementations of DP machine learning.

We review these prior approaches in section 2, and introduce key DP concepts in section 3. Then we present our framework in section 4, followed by results and discussion in section 5. Finally we conclude in section 6.

## 2   Related work

Although differential privacy of a mechanism is established with mathematical proof, previous work has shown that holes can exist, either in theory, such as the sparse vector mechanism (of which erroneous versions have been proposed [10]), or implementation, for example sampling non-uniformity in pseudo-random number generators [15]. Such occurrences point to the need for empirical verification of the promised security of differential privacy mechanisms. This is critical as lapses in DP are likely to provide greater predictive performance, so choosing a model to obtain maximal accuracy given a chosen level of privacy is likely to select these errant models.

Recent works propose efficient solutions for auditing simple differential privacy mechanisms operating on scalar or vector inputs [10, 11]. In such cases, the neighboring inputs can be enumerated tractably (e.g. for a vector input, trying adding one to each input element). For each neighboring pair, the

appropriate output set is determined, and Monte Carlo probabilities are measured to determine privacy. The sampling process in such mechanisms is vectorized to greatly reduce the runtime [11]. These works develop a rigorous statistical framework for DP auditing which serves as the basis of current ML auditing approaches, including our work.

Few studies have attempted to audit DP machine learners. We are aware of two works that measure the privacy of neural networks trained with the DP-SGD mechanism [16]. Jagielski et al. [14] develops a poisoning attack (ClipBKD) by constructing a data point along the axis of least variance in the dataset. This causes the gradients of the point to be distinguishable, mitigating the effect of the DP-SGD clipping and noise. Since this attack is model-agnostic, we adopt it as a baseline for all our models. We show considerably improved performance with our method and that the least variance approach suffers as the sphericity of the dataset increases.

The second work analyzes DP-SGD through a series of increasingly white-box attacks [9]. They note that while the DP-SGD privacy guarantee is formulated for all neighboring datasets, the mechanism itself operates purely via gradient manipulation. Therefore they are able to formulate stronger attacks which directly target the gradient or make use of adaptive poisoning. While these are not usable for general-purpose ML auditing, we implement their poisoning attack, based on adversarial perturbation, for our DP-SGD experiments. We find that this attack performs comparably with ClipBKD on DP-SGD.

While their works are important progenitors to ours, they estimate the privacy loss by thresholding a single scalar output, e.g. the loss on a specific poisoning point. This is a major bottleneck limiting the ability to audit other machine learning algorithms. Instead of a scalar, our approach learns distributions over any vector representation of a model, allowing us to be the first to propose a general procedure for auditing any DP learner.

Another predecessor established the complementary relationship between data poisoning and differential privacy, using a poisoning attack based on gradient ascent of logistic regression parameters $\theta$ with respect to $x$ to reach a target $\theta^\star$ [17]. The result has similar form to a perturbation influence function which we discuss [18, 19]. However, the aim of their study is to evaluate the effect of differential privacy on poisoning strength rather than vice versa. In addition, their attack formulation requires manual gradient computation which does not readily extend to other models. We include a form of their attack to compare in our logistic regression experiments.

## 3 Differential privacy overview

We provide a background on key elements of differential privacy relevant to our work.

**Definition 3.1.** *A randomized learner $\mathcal{M} : \mathcal{D} \times \mathcal{B} \mapsto \Theta$ is a function mapping a training dataset $D \in \mathcal{D}$ and auxiliary noise $b \in \mathcal{B}$ to an output model $\theta \in \Theta$.*

If we train the learner repeatedly on the same dataset, we can obtain a different model $\theta$ each time based on the noise source $b$. For simplicity we implicitly denote the learner as a randomized function $\mathcal{M}(D)$ with a probability distribution $\mathbb{P}$ induced by $b$.

**Definition 3.2.** *A randomized learner $\mathcal{M}$ is considered $(\varepsilon, \delta)$-differentially private if for all datasets $D, D' \in \mathcal{D}$ differing in a single entry and measurable sets $S$ over the output space $\Theta$,*

$$\mathbb{P}(\mathcal{M}(D) \in S) \leq e^\varepsilon \mathbb{P}(\mathcal{M}(D') \in S) + \delta \tag{1}$$

In this definition, the $\varepsilon$ term is the primary quantity of interest because it bounds the ratio between model output probabilities when the original dataset $D$ is perturbed by a single row. Setting $\varepsilon = \delta = 0$ would enforce that $M(D)$ equals $M(D')$ in distribution, while a large $\varepsilon$ or $\delta$ permits the distributions to be arbitrarily different. As $\varepsilon \to \infty$ the model distribution converges to $\mathcal{M}_\infty(D)$, which in most cases is identical to the standard non-private version of the model. The $\delta$ term permits some additive slack in the probability bound, but in most purposes is set to an $o(1/n)$ quantity.

Given $\varepsilon$ and $\delta$, $\mathcal{M}$ calibrates the noise distribution to the *sensitivity* of the model function [20].

**Definition 3.3.** *The sensitivity of a function $f : \mathcal{D} \to \mathbb{R}^d$ is $\max_{D, D'} \|f(D) - f(D')\|$ where $D, D'$ differ by one row.*

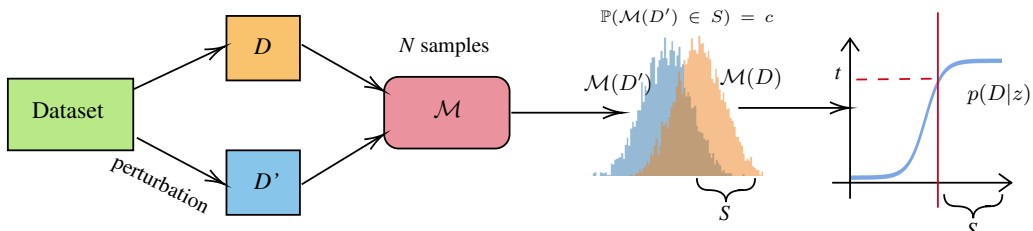

Figure 1: Auditing procedure: A given dataset is perturbed to get $D'$. Mechanism $\mathcal{M}$ is trained on the $D$ and $D'$ to generate $N$ samples each. A classifier is trained on the samples to learn the posterior probability of dataset $D$ conditional on mechanism output $z$. A threshold $t$ is determined based on false positive constraint $c$ to determine the output set $S$.

Our main goal is to audit the privacy of a known mechanism whose inner workings may be hidden, so we assume in our work that the user has access to the mechanism, the privacy parameters, and the dataset, but does not have control over the training process or the randomness $b$.

The final tool which we will employ in our framework is *group privacy*, which iterates over Eq. 1 to bound the probabilities when $D, D'$ differ by multiple points.

**Lemma 3.4.** *Suppose $\mathcal{M}$ is $(\varepsilon, \delta)$-private and $D, D'$ differ by $k$ rows. Then for all measurable $S \subset \Theta$,*

$$\mathbb{P}(\mathcal{M}(D) \in S) \le e^{k\varepsilon}\mathbb{P}(\mathcal{M}(D') \in S) + \delta \cdot \frac{1 - e^{k\varepsilon}}{1 - e^{\varepsilon}} \tag{2}$$

## 4 Auditing framework

A DP mechanism calibrates its randomization to a user-specified $\varepsilon_{th}$ representing the desired theoretical level of privacy. The goal of DP auditing is to empirically assess the actual privacy enabled by the mechanism, $\varepsilon^\star$. To do this, we determine a lower bound for $\varepsilon^\star$ with high statistical confidence. Observe that rewriting the definition of DP gives

$$\varepsilon^\star \ge \ln \frac{\mathbb{P}(\mathcal{M}(D) \in S) - \delta}{\mathbb{P}(\mathcal{M}(D') \in S)}$$

For any given $(D, D', S)$ we can estimate the quantity with Monte Carlo by retraining $\mathcal{M}$ to generate samples and then compute a 95% confidence interval. The lower bound of the interval $\hat{\varepsilon}_{lb}$ is the detected privacy violation, and with high probability, we state that $\varepsilon^\star \ge \hat{\varepsilon}_{lb}$ with $(D, D', S)$ as a witness. By maximizing $\hat{\varepsilon}_{lb}$ over all possible witnesses, we increase the lower bound on $\varepsilon^\star$ which we can then compare against the promised $\varepsilon_{th}$. Given a mechanism $\mathcal{M}$, the auditing objective is then to find

$$\sup_{D,D'} \sup_{S} \ln \frac{\mathbb{P}(\mathcal{M}(D) \in S) - \delta}{\mathbb{P}(\mathcal{M}(D') \in S)}$$

that is, the maximum privacy loss ratio over neighboring datasets $D, D'$ and measurable output set $S$. For all mechanisms considered in this work $\delta < 10^{-4}$ which is not detectable, so we simplify by setting $\delta = 0$.

A successful auditing approach requires solutions to the two separate maximizations. We maximize over $D, D'$ by developing a toolkit of poisoning attacks which have the largest influence on the mechanism in question. Once $D$ and $D'$ are obtained, we fit a likelihood ratio-based optimization to determine an appropriate $S$. In the following, we discuss these procedures in detail.

### 4.1 Optimizing $D, D'$

Given a dataset $D = (X, Y)$ we define a valid poisoning attack to be an operation which selects an existing data point $(x, y)$ and perturbs $x$ to any new point $x^\star$ within a constraint set $\mathcal{C}$. As we consider classification algorithms in this work, the class label $y$ can also be switched. The constraint $\mathcal{C}$ is specific to each DP algorithm and is used to determine the max sensitivity bound from Def. 3.3 (as otherwise a single data point can arbitrarily affect the model). Without loss of generality, we set the constraint to be the smallest containing the training data.

This is the only part of the framework which needs user design, as a strong attack against one mechanism may not be effective against another. However we distill the steps for constructing such attacks into a recipe:

1. The underlying model of $\mathcal{M}$ can be abstract, requiring a summary function $\tau$ to embed $\mathcal{M}$ into a vector space. Since post-processing of DP outputs does not decrease privacy [6], we can select $\tau$ to preserve as much information about $\mathcal{M}$ as possible. In logistic regression the transformation is natural: we define $\tau \circ \mathcal{M}$ as the coefficient vector $\theta$ (Table 1). To reduce notation overload we implicitly apply $\tau$ when we discuss empirical samples $z \sim \mathcal{M}(D)$.

2. Obtain a non-private algorithm $\mathcal{M}_\infty(D)$ as a surrogate. Note that as long as $E_b[\mathcal{M}(D, b)] \approx \mathcal{M}_\infty(D)$, meaning the DP model is on average equivalent to the non-private version, it is sufficient to find the optimal poisoning with respect to the non-private model. While not explored in this work, attacks can instead take into account the actual noise distribution by averaging over samples of $\mathcal{M}(D, b)$, at the cost of additional run-time.

3. Determine a poisoning $(x^\star, y^\star)$ from $(x, y)$ maximizing the distance between $\tau(\mathcal{M}_\infty(D))$ and $\tau(\mathcal{M}_\infty(D'))$. This involves a selection and a perturbation step.

| | Bound $\mathcal{C}$ | Summary $\tau$ |
|---|---|---|
| Log. reg. | $L_2$-norm | coefficient $\theta$ |
| NB | hyper-rectangle | $\mu_{yd}, \sigma^2_{yd}, \pi_y$ |
| RF | hyper-rectangle | $\Pr(x^\star)$ of each tree |
| DP-SGD | $L_2$-norm of $\nabla_\theta L(x, y; \theta)$ | $\Pr(x^\star), \Pr(\vec{0}), \Pr(x_i^{test})$ |

Table 1: Constraints and summary functions used.

Ahead we detail the poisoning attack design for each mechanism:

**Logistic Regression.** DP logistic regression is achieved using one of three approaches: objective, output, or gradient perturbation. We evaluate objective and output perturbation using the widely used method developed by Chaudhuri and Monteleoni [21] and Chaudhuri et al. [22].

For the attack we adopt the influence function, a technique which estimates the change in a model when a specific data point is infinitesimally upweighted in the training set [19, 23]. For a general model with gradient and Hessian information, the effect of point $x^\star$ on the model parameters for a loss function $L$ can be approximated as $\mathcal{I}_\theta(x^\star) = -H_\theta^{-1} \nabla_\theta L(x^\star, \hat{\theta})$. In generalized linear models with canonical link function where $L$ is the negative log likelihood, this has a closed form involving the Fisher information as the expected Hessian [18], which for logistic regression evaluates to $\mathcal{I}_\theta(x^\star) = (y^\star - \hat{y}^\star)(X^\top W X + \lambda I)^{-1} x^\star$ where $W$ is a diagonal matrix with entries $W_{ii} = \hat{y}_i(1 - \hat{y}_i)$, and $\lambda$ regularization.

We choose $\tau$ to be the coefficient vector $\theta, \theta'$ of $\mathcal{M}_\infty(D), \mathcal{M}_\infty(D')$ respectively. We first select $(x, y)$ closest to the corner of the hyper-rectangle containing the data (a heuristic for selecting a point far from where the separating hyperplane would likely be) and initialize $(x^\star, y^\star)$ as the mean point in the opposite class. Since our goal is to increase the distance $\|\theta - \theta'\|_2$, we optimize the $L_2$ norm of the influence function using projected gradient ascent. The constraint set $\mathcal{C}$ is the $L_2$ norm ball containing the training set, so after each iteration we clip $x^\star$ if needed.

**Naive Bayes.** The Gaussian Naive Bayes model parameterizes the class-conditional distribution of features using independent Normal distributions. The parameters involve a mean $\mu_{yd}$ and variance $\sigma^2_{yd}$ for each feature and class combination, as well as prior probabilities for each class $\pi_y$. To achieve differential privacy, Laplace or Geometric noise are added to the maximum likelihood estimates for each parameter [12]. The constraint set $\mathcal{C}$ is a hyper-rectangle set by the smallest and largest value of each feature in the dataset.

We choose $\tau$ to be the vector of all $\{\mu_{yd}, \sigma^2_{yd}, \pi_y\}$. The maximum influence a perturbation can have on $\hat{\mu}_y$ of class $y$ is by selecting a point on the corner of the hyper-rectangle $\mathcal{C}$ nearest the data points of class $y$ and placing it on the corner furthest away. On the other hand, the maximum influence attack on $\pi_y$ is to flip a class label. We combine the two ideas by flipping the class label of the point closest to a corner. In our experiments we compare the power against only attacking $\mu_y$.

**Random Forest.** The DP random forest mechanism uses unsupervised random splits of the features based on the domain of the inputs. Each tree is split to a pre-determined depth. Then the training points are percolated through the forest, and the majority label of each leaf is sampled from the Exponential mechanism [13]. Thus a perturbed training point has no impact on the actual tree structure besides the potential label of a single leaf in each tree. As a result we measure only the change in those leaves, choosing $\tau$ to be the prediction of each tree on $x^\star$.

The most detectable change in probability occurs when $j = 1$ and then the majority class is swapped by a class flip. For example, if a leaf has one positive and two negative points, we flip the class of one of the negatives. Since each tree is random, there is no guarantee that any given point will be in such a situation where $j = 1$, but to increase this chance we target points which are likely to be solitary in each leaf. Thus we select the point most distant from the rest of the training set as measured by $L_1$-distance and flip its label. We provide further exposition and hyperparameter details in the Appendix.

**DP-SGD.** For DP-SGD we evaluate two extent poisoning attacks using our framework. The *ClipBKD* attack constructs a data point along the axis of least variance in the dataset [14]. This can be performed by taking the eigenvector of the covariance matrix with the smallest eigenvalue, and then projecting it to the median L2 norm of the training set. This point is assigned the label with smallest predicted probability. The goal of this attack is to enable the gradients of the point to be distinguishable, mitigating the effect of the DP-SGD clipping and noise.

The second attack selects a random datum and maximizes the loss with gradient ascent. The loss is defined with respect to a set of trained shadow models [9]. In both attacks, the outcome $\tau(\mathcal{M}(D))$ being measured is the predicted probability of the perturbed point $P_\mathcal{M}(x^\star = 1)$, while ClipBKD first subtracts the zero vector's predicted $P_\mathcal{M}(\vec{0})$. We also consider an updated ClipBKD with the zero prediction encoded separately with additional test predictions in $\tau$: $P_\mathcal{M}(\vec{0})$, $P_\mathcal{M}(x_i^{test})$.

### 4.2 Optimizing $S$

Our approach for determining $S$ is based on the framework of Bichsel et al. [11], which derives the optimal region for distinguishing two vector distributions. Given candidate datasets $(D, D')$ with DP distributions $\mathcal{M}(D)$ and $\mathcal{M}(D')$, we seek $S$ maximizing $\mathbb{P}(\mathcal{M}(D) \in S)/\mathbb{P}(\mathcal{M}(D') \in S)$. Let $f_{\mathcal{M}(D)}$ and $f_{\mathcal{M}(D')}$ be their respective densities. Selecting $S$ to best distinguish two distributions is analogous to identifying an optimal rejection region in hypothesis testing. From the Neyman-Pearson Lemma (cf. Appendix), the maximum power test at level $c$ is defined by rejection region $S := \{z \mid f_{\mathcal{M}(D)}(z)/f_{\mathcal{M}(D')}(z) > k\}$ for some $k > 0$, where $\mathbb{P}(\mathcal{M}(D') \in S) = c$ [24].

Intuitively, we have a likelihood ratio $f_{\mathcal{M}(D)}(z)/f_{\mathcal{M}(D')}(z)$ and we are selecting points to fill $S$ where $D$ is more likely than $D'$. We start by adding points where $D$ is most likely and work downwards until $S$ is large enough that there is a $c$ chance that $\mathcal{M}(D')$ is erroneously in $S$.

Directly obtaining these densities is intractable. Instead we can learn, given collected samples $\{\mathcal{M}(\mathbf{D})\}$ where $\mathbf{D} \in \{D, D'\}$, the posterior probability of the dataset $D$:

$$p(D|z) := \Pr(\mathbf{D} = D | \mathcal{M}(\mathbf{D}) = z)$$

Let us consider the set $L^k := \{z \mid p(D|z)/p(D'|z) > k\}$. From Bayes' Theorem we know $p(D|z) \propto f_{\mathcal{M}(D)}(z) \cdot \Pr(\mathbf{D} = D)$. Given equal samples of $\mathbf{D}$ from $D$ and $D'$, then $P(\mathbf{D} = D) = P(\mathbf{D} = D')$ implying $p(D|z)/p(D'|z) = f_{\mathcal{M}(D)}(z)/f_{\mathcal{M}(D')}(z)$.

Therefore $S = L^k$ and we use it as the rejection set. Finally, we can further simplify $S$ by observing that $p(D|z) + p(D'|z) = 1$, and so $S = \{z \mid p(D|z) > t\}$ where $t := k/(1 + k)$. What remains is to set $t$ according to the level $c$ so that $\mathbb{P}(\mathcal{M}(D') \in S) = c$. As this is equivalent to $\mathbb{P}(p(D|z) > t \mid z \sim \mathcal{M}(D'))$, we set $t$ empirically to correspond to the $(1 - c)$-quantile of $\mathcal{M}(D')$.

Our approach introduces the following further adaptations on Bichsel et al. [11] to enable auditing on slower, potentially expensive machine learning mechanisms. (The original work used around $10^8$ samples for auditing simple mechanisms, which is infeasible for training machine learning models. We use $N = 10000$ samples for all mechanisms except DP-SGD where $N = 500$.)

1. We make the method flexible by searching $c$ empirically to maximize $\tilde{\varepsilon}_{lb}$, rather than setting $c = 0.01$ as in the original. When $N \to \infty$, smaller values of $c$ are optimal since the likelihood ratio is largest. At smaller sample sizes, however, this is offset by increasing uncertainty in Monte Carlo estimation of small probabilities.

---

**Algorithm 1** ML-Audit: Optimizing output set $S$

---

**Require:** Datasets $D, D'$, DP mechanism $\mathcal{M}$, min. probability constraint $r$, confidence $\alpha$, sample size $N$
1: Sample $z_1, \ldots, z_N \sim \mathcal{M}(D), z'_1, \ldots, z'_N \sim \mathcal{M}(D')$
2: Generate labels $y_i = 1$ if $z_i$, else $y_i = 0$ if $z'_i$
3: Train classifier on $\{z, z'\}, \{y\}$ to learn $p_\theta(D|z)$
4: $p_1, \ldots, p_N \leftarrow p_\theta(D|z_1), \ldots, p_\theta(D|z_N)$
5: $p'_1, \ldots, p'_N \leftarrow p_\theta(D|z'_1), \ldots, p_\theta(D|z'_N)$
6: $\tilde{\varepsilon}_{lb} \leftarrow -\infty; \tilde{t} \leftarrow 0$
7: **for** $t$ in sort($\{p_i\}_1^N \cup \{p'_i\}_1^N$) **do**
8: $\quad n_1 \leftarrow \sum_i \mathbb{1}(p_i > t)$
9: $\quad n_0 \leftarrow \sum_i \mathbb{1}(p'_i > t)$
10: $\quad$ **if** $n_0/N < r$ **then** $\hfill \triangleright$ Avoid estimating probs $< r$
11: $\quad\quad$ skip to next iter
12: $\quad \varepsilon_{lb}^t \leftarrow \text{Eps\_LB}(n_1, n_0, N, \alpha/2)$ $\hfill \triangleright$ Lemma A.3
13: $\quad$ **if** $\varepsilon_{lb}^t > \tilde{\varepsilon}_{lb}$ **then**
14: $\quad\quad \tilde{\varepsilon}_{lb} \leftarrow \varepsilon_{lb}^t$
15: $\quad\quad \tilde{t} \leftarrow t$
16: **return** $S \leftarrow \{z \mid p_\theta(D|z) > t\}$

---

2. We use group privacy to increase statistical distinguishability, a trick used also in Jagielski et al. [14]. After obtaining $(x^\star, y^\star)$, we insert $k$ copies of the point. From Lemma 3.4 the detected privacy $\tilde{\varepsilon}_{lb}$ is reduced to $\tilde{\varepsilon}_{lb}/k$. However, the extra copies causes $\mathcal{M}(D')$ to be more distinguishable which makes $\mathbb{P}(\mathcal{M}(D) \in S)$ larger relative to $\mathbb{P}(\mathcal{M}(D') \in S)$. This is also a tradeoff: At small true $\varepsilon$ the lower bound $\tilde{\varepsilon}_{lb}$ improves despite division by $k$, whereas at high $\varepsilon$ the distributions are sufficiently different at $k = 1$ that it is not worth dividing $\tilde{\varepsilon}_{lb}$. We include empirical analysis of $k$ in our Appendix.

3. All previous auditing works (including below) use Clopper-Pearson intervals for $\hat{p}_1$ and $\hat{p}_0$ to determine the lower bound. This method is highly suboptimal because it separately bounds the numerator and denominator. We identify instead the Katz-log confidence interval to directly bound the ratio of binomial proportions [25], see Lemma A.3. This has far better coverage properties in our simulations, as shown in the Appendix.

Our modified procedure is presented in Algorithm 1 and summarized visually in Fig. 1.

**Comparison to other approaches.** While our method allows $\mathcal{M}(D)$ to be any vector of information, the previous works auditing DP-SGD require $\mathcal{M}(D)$ to be a single value (e.g. prediction probability or loss) and search for a threshold dividing the score into $S$ and $S^c$ [9, 14]. Our method can reduce to this special case by choosing $\tau$ to map to a scalar. The approach can be interpreted as membership inference with the bound $FN + e^\varepsilon FP \leq 1 - \delta$ from Kairouz et al. [26], where $\{\mathcal{M}(D) \in S\}$ are true positives and $\{\mathcal{M}(D') \in S\}$ are false positives. These works also note that flipping the definition of false positive and false negatives enables testing the complement $\mathbb{P}(\mathcal{M}(D') \in S^c)/\mathbb{P}(\mathcal{M}(D) \in S^c)$. As in those works we select $S^c$ instead of $S$ if it gives higher $\tilde{\varepsilon}_{lb}$.

**Corollary 4.0.1.** *Given Algo. 1 and $N$ samples, the highest detectable privacy risk is*

$$\ln(N) - z_{\alpha/2}\sqrt{1 - \frac{1}{N}}$$

**Estimating** $\hat{\varepsilon}_{lb}$**.** After $(D, D', S, k)$ are determined, the final step is to draw $N$ fresh samples of $M(D)$ and $M(D')$ for an independent Monte Carlo verification. As in the search phase, we compute $n_1 = \sum_i \mathbb{1}(M(D) \in S)$, $n_0 = \sum_i \mathbb{1}(M(D') \in S)$ and use the lower bound of the Katz log interval to return our final estimate $\hat{\varepsilon}_{lb}$.

To avoid biasing the confidence interval, it is important that the final MC estimate is conducted on an independent set of samples, as done in [11, 14]. This is especially the case when optimizing over thresholds in $S$. Since we compute a $1 - \alpha$ interval over $\hat{\varepsilon}$ for each threshold, using the best result from the same sample would lead to biased inference from multiple testing.

# 5 Experiments and Results

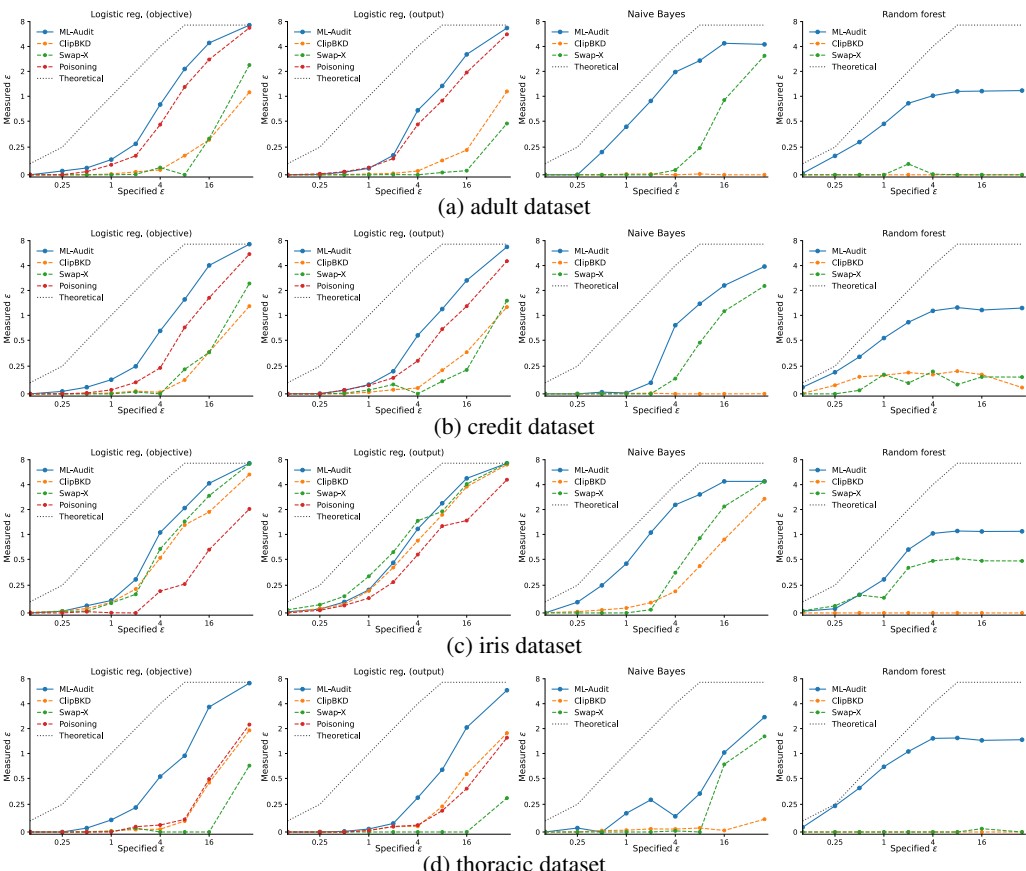

(a) adult dataset

(b) credit dataset

(c) iris dataset

(d) thoracic dataset

Figure 2: Specified (x-axis, log-scale) vs detected privacy risk (y-axis, log-scale) using our ML-Audit framework comparing our perturbation attack (blue), ClipBKD (orange) and a row-swap baseline (green), over four datasets. For logistic regression, an additional poisoning attack (red) can be considered an ablation of our approach. Remaining datasets in Appendix Figure 5. In highly non-spherical datasets such as iris, row-swap performs comparably to ML-Audit on logistic regression, which we discuss in 5.

We evaluate the DP mechanisms over a range of $\varepsilon_{th}$: $\{0.1, 0.25, 0.5, 1, 2, 4, 8, 16, 50\}$. At $\varepsilon_{th} = 0.1$ the ratio of probabilities is bounded by $e^{0.1} \approx 1.1$ giving nearly indistinguishable distributions, whereas at $\varepsilon_{th} = 50$ essentially no privacy is guaranteed. For a given dataset $D$, we perturb $k \in \{1, 2, 4, 8\}$ points to get $D'$ and train $N = 10000$ times for each to determine the appropriate auditing set $S$. Then we obtain $N$ new samples to perform the final Monte Carlo estimate and obtain the lower bound $\hat{\varepsilon}_{lb}$. We use confidence level $\alpha = 0.05$ throughout.

We assess Naive Bayes, logistic regression (output and objective perturbation), and random forest on common machine learning datasets: *adult*, *credit*, *iris*, *breast-cancer*, *banknote*, *thoracic*. We use the diffprivlib library [27] and implement output perturbation following [22]. Additionally, we test DP-SGD on FMNIST and CIFAR10 (here with $N = 500$) using [28]. Refer to Appendix for more details.

Figure 2 directly compares, over three datasets, the $\hat{\varepsilon}_{lb}$ obtained by our poisoning attacks (ML-Audit), the ClipBKD attack, and a baseline where a random data point $x$ is changed to a random point from the opposite class without changing labels (Swap-X). (Note that while the original ClipBKD work uses DP-SGD, the poisoning itself is mechanism-agnostic, so it is an appropriate baseline.) We replicated our procedure 3x and show the median $\hat{\varepsilon}_{lb}$. The Katz log interval was used for the lower bounds even on the baselines. As a reference we plot the theoretical privacy loss up to the maximum detectable limit (Corollary 4.0.1).

For the logistic regression experiments we include an additional alternative of our approach where the modified point, rather than maximizing the influence, is optimized to target a specific coefficient vector (labeled Poisoning). The method is based on the coefficient attack of [17], a gradient-based optimization approach with similar form to ours. Heuristically, we want the target coefficient vector to be as distinguishable as possible from the original optimal linear coefficients, so we constructed a perpendicular vector to the original coefficients using cross products. This method, which only exists for logistic regression, can be considered an ablation of ML-Audit with a weaker attack.

Our attacks consistently and often dramatically outperform baselines across datasets, models, and $\varepsilon_{th}$. In some cases, the privacy detection is within a small factor $(1-2\times)$ of optimal. The Poisoning attack is close to our attacks in some datasets but performs poorly in others. The Swap-X baseline shows the expected privacy loss of an attack which poisons a data point within the original data distribution, rather than a worst-case perturbation (e.g. swapping two points in the training set). The only case where Swap-X performs similarly to our tailored attacks is in output-perturbed logistic regression on separable datasets with high non-sphericity such as *iris* and *breast-cancer*. We believe these properties enable simple strategies to work well (see Fig. 3a). We believe this is not a meaningful difference in performance as both methods are near the limit of what can be theoretically detected. Our results validate previous findings showing that estimating $\varepsilon$ using membership inference-based bounds are weaker than poisoning attacks [9, 14]. We also tried the loss-threshold attack of [7, 29] and found it ineffective.

On many models and datasets we obtain nearly optimal results, indicating that these datasets are close to the maximal sensitivity of the mechanism. We observe this for random forest at $\varepsilon_{th} \leq 2$ but also a plateau in higher $\varepsilon_{th}$. This is likely due to lack of resolution in the outputs: an average over $m$ tree decisions only takes $m$ unique probability values. Our attack on Naive Bayes is also near-optimal, while varying among datasets. Since the algorithm assigns $\varepsilon_{th}/3$ to protect each of $\mu_{yd}$, $\sigma_{yd}^2$, and $\pi_y$, a perturbation needs to achieve the sensitivity bound on all three to reach the theoretical privacy risk, with the bound over all pairs $D, D'$ rather than conditioning on a starting $D$. For example, to reach the bound for $\mu$, all points of one class must be in one corner of the hyper-rectangle and the poisoned point on the other corner, which is unrealistic. For this reason, our choice of attacking the class prior $\pi$ is generally more effective than attacking the mean $\mu$ (Fig. 4).

Lastly, our DP-SGD experiments assess the two existing perturbation methods of [9, 14]. Our findings (Fig. 8) are consistent with what is reported in those works. Interestingly, we find that the shadow adversarial attack outperforms ClipBKD on FMNIST and vice versa on CIFAR10. We also updated ClipBKD to use additional information in $\tau$ as detailed in Section 4.1 and find a moderate improvement on FMNIST. Refer to the Appendix for further discussion.

**Dataset-specific attack comparisons.** Previous work hypothesized that privacy attacks on realistic datasets do not cause the model shift to reach the worst-case given by the sensitivity bound, except when the attacker can completely specify the dataset [9]. Based on our observations on the impact of the dataset distribution on attack strength, our evidence supports the hypothesis that achievable privacy is dataset-dependent. We believe theoretical analysis of datasets for attack risk to be useful for future work.

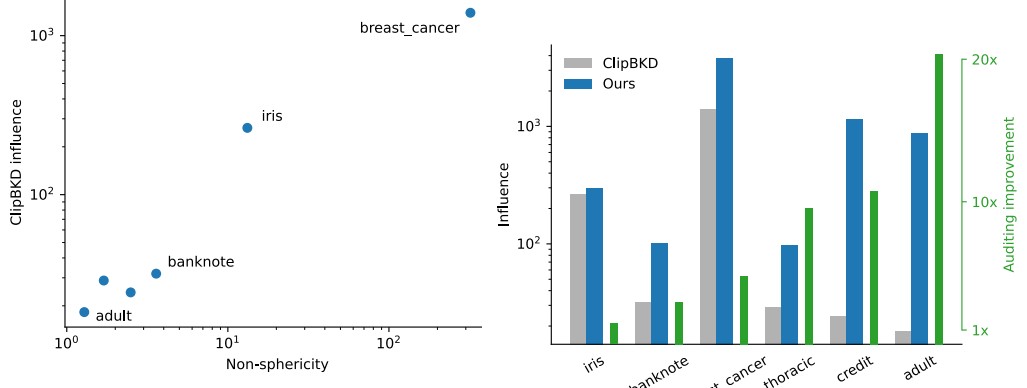

Figure 3: (a) Non-sphericity vs ClipBKD influence (b) Influence of ClipBKD perturbation compared to our perturbation explains the difference in performance across datasets.

We briefly investigate this phenomenon by comparing the performance of ClipBKD by dataset sphericity, as computed by ratio between largest and smallest singular values. An attack on the direction of least variance is likely ineffective when the dataset is spherical. We observe a strong relationship between the non-sphericity of a dataset and the influence of the ClipBKD perturbed point in Fig. 3a. In comparison our influence-based attack consistently finds a higher influence value (Fig. 3b). Furthermore, this difference in influence is directly linked to the auditing improvement of our approach compared to ClipBKD.

**Detecting violations in privacy**. We discuss two case studies where privacy leaks occur in practice and are detected by our method. First, the Naive Bayes mechanism in *diffprivlib* exposes perturbed class counts in the API. However, the sum of class counts is enforced to be the dataset size. This means whenever $(D, D')$ differ by one row in length no privacy is guaranteed. When we include the exposed class counts as part of $\tau$, our framework detects maximal privacy loss at all $\varepsilon_{th}$ (Fig. 4a). We therefore recommend that only the class priors and not counts be made accessible.

As another example, in some works DP defines neighboring datasets can only have a single row addition or deletion, while others allow modifiying an existing point (equivalently deleting and then adding a row). This means that depending on the implementation, the actual privacy level may be half or double what the user desires to obtain. Our auditing framework correctly detected this discrepancy in the DP random forest, which defines $\varepsilon$ under the first option. Our estimated $\hat{\varepsilon}$ using the second definition exceeded the theoretical level until the definitions were reconciled.

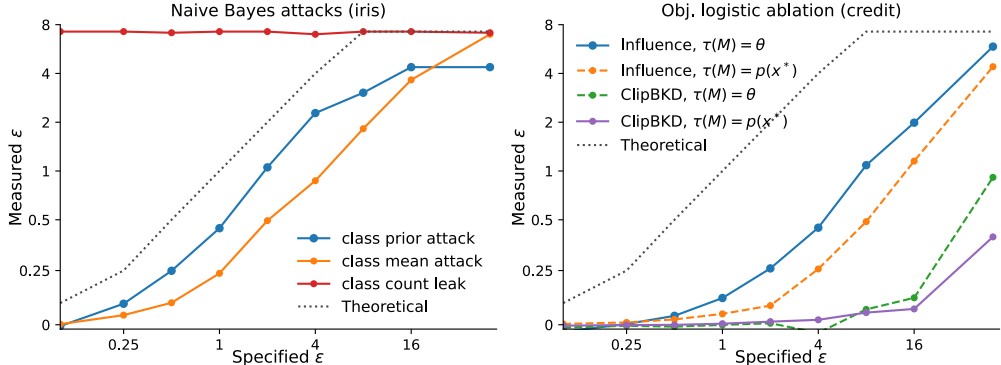

Figure 4: (a) Naive Bayes: Optimal attack on $\pi$ (blue) compared to attacking $\mu$ (orange). However, when $D, D'$ differ in length and class counts from the API are obtained, all privacy is compromised (red). (b) Logistic regression ablation showing incremental improvements when upgrading the baseline ClipBKD (purple) to our approach (blue).

We reiterate that the advantages of our approach over precursors are (1) improved perturbation attacks on $D$, and (2) optimizing a likelihood attack $S$ on any summary function $\tau$ of a model $\mathcal{M}$. Our final ablation in Fig. 4b highlights these strengths by demonstrating the incremental improvement to ClipBKD when either (1) or (2) are added.

## 6   Conclusion

We have proposed ML-Audit, a framework for estimating the differential privacy of a ML model. ML-Audit provides a recipe for devising audits against arbitrary models and is often orders of magnitude more effective than existing approaches – sometimes $\leq 2\times$ of theoretical optimum.

## 7   Acknowledgments

Approved for Public Release; Distribution Unlimited. PA #: AFRL-2022-3247.

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
