# OpenReview forum: "A General Framework for Auditing Differentially Private Machine Learning"
_NeurIPS.cc/2022/Conference — NeurIPS 2022 Accept_

### Official Review · Reviewer_K5aD · 2022-07-07

**Rating:** 6
**Confidence:** 4
**Soundness:** 3 good
**Presentation:** 3 good
**Contribution:** 2 fair

**Summary:**

The paper considers the problem of auditing machine learning models. They propose strategies to improve upon existing machine learning auditing approaches, and design poisoning attacks to audit a wider range of model types than have been previously considered in the auditing literature, such as Naive Bayes, logistic regression, and random forests. They show that their approach can be used both for bug finding and for understanding properties of a dataset which impact empirical privacy.

**Questions:**

Is the influence metric correlated with the gap between theoretical and empirical epsilon across datasets?

Did you ever experiment with baselines besides flipping and ClipBKD, something like standard backdoor attacks or the Nasr et al strategy?

**Limitations:**

This is fine.

**Strengths And Weaknesses:**

Strengths:

Using confidence intervals for the ratio of binomials is a nice improvement.

The paper designs model-specific poisoning attacks that work quite well. DP-SGD is not the only interesting DP learning algorithm, so this is a well motivated problem.

I liked the discussion of the impact of k.

Weaknesses:

The paper overclaims on the novelty of the framework. It is not very different from existing techniques. The main differences are the combination of DP-Sniper auditing with ML poisoning as proposed by Jagielski et al and Nasr et al, as well as the use of binomial ratio confidence intervals. The main contribution of the paper seems to be in the instantiations of the framework rather than the framework itself.

Strange choices of baselines. In the Jagielski et al. paper, ClipBKD is designed specifically for DP-SGD. It's a stretch to use it to audit objective perturbation, let alone Naive Bayes. That paper also used backdoor poisoning attacks. Nasr et al. have an optimization procedure that could be used to get model-adapted poisoning attacks. These are all appropriate baselines. What would be even cooler (and probably a contribution in its own) is taking advantage of the large literature on poisoning attacks (including specifically on objective perturbation and output perturbation for logistic regression [r1], nonprivate Naive bayes [r2]). These are generally not designed for auditing, so an auditing-specific poisoning strategy should still outperform them.

The motivation for dataset-specific privacy does not seem to be supported by the experiments. The authors want to investigate the idea that "achievable privacy is dataset-dependent", but the investigation focuses on a comparison between influence-based poisoning and ClipBKD. The achievable privacy is better measured by something like the maximum epsilon obtained over all attacks, not a comparison between attacks. What might help better make this point is investigating whether the "Ours" influence metric the authors consider in Figure 3b is correlated with the empirical epsilon lower bound, when analyzed across datasets (rather than across attacks on the same dataset). This might help give a cool metric to predict privacy risk before going through the auditing process.

[r1] - https://www.ijcai.org/proceedings/2019/0657.pdf
[r2] - https://people.eecs.berkeley.edu/~adj/publications/paper-files/SecML-MLJ2010.pdf

---

> ### Author Response · Authors · 2022-07-29
> **Initial Response**
>
> **Novelty:** We acknowledge and agree with the reviewer's comments that our work builds on the technical apparatus of DP-Sniper, and we clarify our Introduction and Related Works to make this connection clearer. However we do wish to note that our auditing framework is a substantial improvement over each of the prior works (DP-Sniper, Jagielski et al, Nasr et al.). Our method combine strengths from each of the preceding papers in a non-trivial way (e.g. vector output modeling from DP-Sniper, group privacy from Jagielski et al, threshold search in Jagielski et al and Nasr et al), while also introducing improved confidence intervals and novel perturbation/influence attacks. We believe the often significant improvements we see in auditing, reinforce that our work combining these ideas is non-trivial.
>
> To give some more context: in order to adapt the DP-Sniper framework, we needed solutions to reduced sample size as well as transitioning from auditing simple DP mechanisms to complex ML models. The latter required developing new methods to select and perturb neighboring datasets. While Jagielski and Nasr devised such novel attacks for DP-SGD specifically, we have done the same for three other DP ML models (logistic reg, NB, RF) which had not been attempted. The latter two works also use threshold search which improves over DP-Sniper's fixed threshold for reduced sample sizes, but they did not use a framework which could model distributions over vectors, which we showed is a large disadvantage in our ablations.
>
> Edit: We also wish to emphasize that in the area of auditing estimation itself, principled improvements are relatively small because they must maintain statistical accuracy of a standard procedure. In particular, each of the prior works mentioned (Bischel et al, Jagielski et al, Nasr et al) are themselves relative improvements upon [Ding et al. Detecting Violations of Differential Privacy]. In terms of designing a procedure to audit general ML models (a "more universal" framework as Reviewer 8PAF says), we are the first.
>
> **Baselines:** We agree that ClipBKD is not ideal as a baseline outside of logistic regression and DP-SGD, and we ended up including it simply because of the lack of prior auditing baselines outside of DP-SGD. In Jagielski et al., they additionally test another backdoor poisoning attack which applies a mask to part of an image. The analog for our work would be to set random features to 0, which we do not anticipate would be stronger than the Swap-X baseline (since flipping labels is more adversarial in a sense).
>
> Jagielski et al. also use a standard membership inference (MI) attack as their baseline. We tried MI attack and found it was ineffective for the non-deep learning models. As a result, we developed the swap baseline which performs much better. This baseline also has an intuitive interpretation because it measures the privacy risk of an _average_ data perturbation, while the ML-Audit attack measures the risk of a _worst-case_ perturbation. While Nasr et al. develop model-adapted poisoning attacks, they are specific for DP-SGD; doing something similar for other ML models was essentially our goal for developing our own model-specific perturbations. In a sense we wish our perturbation attacks to be the baseline which future scientists can improve upon.
>
> *Additional baselines:* Following your suggestion, we have added the additional baseline for logistic regression based on Ma et al. [r1]. As you hypothesized, the attack is not as strong as the ML-Audit attack, but it tends to outperform the swap baseline. The attack is actually very similar to the influence attack, being based on 2nd-order approximation of the loss function, but with a different target coefficient, as discussed in our Related Works. For the adversarial target coefficient we use a perpendicular vector to the true model coefficient. We also looked at the spam Bayes attacks from [r2] as you suggested. To our understanding these are focused on text representations -- after converting the text attack to numerical features, they would be of similar fashion to inserting a poisoning point of the wrong class (essentially swap-X)
>
> Your notes on Dataset-specific privacy are valuable, and we are conducting some additional tests before we finish this part of the response. We will post these new tests once they are complete, and appreciate your patience.
>
> **Update:** We updated the PDF with the appendix attached, and placed new figures at the end corresponding to the Ma et al. baseline (labeled Poisoning) and also the dataset-specific epsilon analysis. While there does not appear to be a clear relation between perturbation influence and the estimated-to-true epsilon gap, we believe this analysis to be interesting nonetheless and highly appreciate your suggestions. We agree our previous figures were not focused on dataset-specific privacy, so we will clarify that in the paper.

---

> > ### Comment · Reviewer_K5aD · 2022-08-08
> > **Thank you for the replies**
> >
> > Thank you for your responses! A couple comments:
> >
> > I'm impressed you implemented the Ma et al. attack! Is there a reason to not put these plots in the main body of the paper (i.e. as a replacement for Figure 2)?
> >
> > Re: Backdoor poisoning would set features to 0
> > Typically, backdoor poisoning also involves a label flip. I would expect this to perform stronger than Swap-X. This is also what you'd do in the [r2] attack.
> >
> > Re: "While Nasr et al. develop model-adapted poisoning attacks, they are specific for DP-SGD; doing something similar for other ML models was essentially our goal for developing our own model-specific perturbations."
> > I think this is a nice way to relate the two works - you should add something like this in the Nasr et al related work paragraph. I think it is at least worth mentioning somewhere why you used ClipBKD, which is designed for DP-SGD, as a baseline, but not Nasr et al, which is also designed for DP-SGD.
> >
> > Re: "threshold search in Jagielski et al and Nasr et al"
> > I think this is worth mentioning in line 265.
> >
> > Re: dataset-specific privacy
> > I'm not sure "relative to the average influence norm of the dataset" is the exact right thing to look at here, since there's going to inherently be a large variance in these influence norms. There are a couple things that I personally think are interesting here to look into: dataset size, dimension, some measure of how concentrated the examples are. In any case, I don't think it's too big a deal if a preliminary look at this doesn't turn up too much, but I would recommend making the clarification on dataset-specific privacy.
> >
> > I'm increasing my score.

---

> > > ### Author Response · Authors · 2022-08-09
> > > **Response to comments**
> > >
> > > Thank you for your suggestions and your interest in the results!
> > >
> > > Re: Fig 2, we will put the new attack(s) in the main body, we were just thinking that it would be more convenient at first.
> > >
> > > Re: backdoor poisoning, thank you for explaining. We will try those attacks out. In Swap-X, we swap the features to a point of the opposite class while keeping the original label, so that is approximately equivalent to flipping the label of that point. So indeed further perturbing the features at that point could cause an improvement.
> > >
> > > Re: adding to the manuscript, we agree and will add your suggestions. The key reason for using clipBKD rather than Nasr et al. attacks was that clipBKD is the most model-agnostic of the prior approaches, being PCA-based. The Nasr et al. attacks are more deep learning-specific e.g. modifying gradients or points on separate training iterations, or crafting an adversarial point w.r.t. cross-entropy loss, so we thought they would make less sense as a baseline. We will explain this in the manuscript as well.
> > >
> > > Re: dataset-specific privacy. We will try out your suggestions, and regardless will change the 'dataset-specific privacy' part in the main body to 'dataset-specific attack comparison'.
> > >
> > > We highly appreciate your insights! We expect the new experiments will run until later this week. If OpenReview lets us we will update again then - otherwise please be assured we will include your valuable suggestions in a (hopeful) camera ready.

---

### Official Review · Reviewer_8PAF · 2022-07-10

**Rating:** 6
**Confidence:** 3
**Soundness:** 3 good
**Presentation:** 3 good
**Contribution:** 3 good

**Summary:**

The paper proposed ML-Audit, a framework for estimating the differential privacy of a ML model. It is a general methodology to empirically evaluate the privacy of differentially private machine learning implementations, combining improved privacy search and verification methods with a toolkit of influence-based poisoning attacks. ML-Audit provides a recipe for devising audits against arbitrary models and is often orders of magnitude more effective than existing approaches.

**Questions:**

Nothing to ask

**Ethics Review Area:**

["I don’t know"]

**Limitations:**

Nothing

**Strengths And Weaknesses:**

Strengths:
1. The idea of proposing a framework for estimating the differential privacy of an ML model is quite novel and significant, as it is more universal with qualitatively improved performance.

2. The experiment session is solid, since this paper assesses Naive Bayes, logistic regression, and random forest on 6 common machine learning datasets and is compared with several strong baselines.

3. DP-based machine learning methods are theoretically rigorous approaches that provide a worst-case privacy guarantee for ML algorithms, which would have broad application prospects in many aspects, such as being used for practitioners to estimate the actual risk they incur in practice.

Weaknesses:
1. We suggest adding deep learning methods as the base models to the experiment, like DP-SGD, mentioned in line 34. Applying the proposed methods to deep learning models makes the paper more persuasive and valuable.

2. In Figure 2 (c), some results of OURS(ML-Audit) do not excel Swap-X baseline, and Swap-X performs similarly to OURS(ML-Audit) in output-perturbed logistic regression on linearly separable datasets such as iris. It will be better to give a corresponding explanation for such a case.

---

> ### Author Response · Authors · 2022-07-29
> **Response to 8PAF**
>
> 1. While we did not train deep learning directly on the 6 standard datasets since they tend to be small, we do have separate DP-SGD results on FMNIST and CIFAR10 in the Appendix. Since prior work focused on DP-SGD, our main focus was bringing attention to the usefulness of auditing other ML models in a general framework. We did replicate the Jagielski et al/Nasr et al attacks and assessed the impact of modifying the Jagielski attack to use a vector output with ML Audit, with moderate improvements in FMNIST (Appendix Fig. 8).
>
>
> 2. There is one instance where Swap-X performs incrementally better than ML-Audit, yes. We believe this is not a meaningful difference in performance as both methods are near the limit of what can be theoretically detected.  This occurs because the iris dataset has very high non-sphericity, which allows simple strategies to work well. For example Figure 3 (a) shows the breast_cancer dataset also has high non-sphericity, and against in Figure (5) we see Swap-X and ML-Audit approach the theoretical limit. This shows that the closeness is an artifact of these datasets, and we will make this clearer in the manuscript and in the caption.
>
> We hope this satisfies your concerns, please let us know if you have any other questions!

---

### Official Review · Reviewer_7BRu · 2022-07-25

**Rating:** 6
**Confidence:** 3
**Soundness:** 3 good
**Presentation:** 3 good
**Contribution:** 3 good

**Summary:**

The paper proposes a general methodology to empirically evaluate the privacy of differentially private machine learning implementations, combining improved privacy search and verification methods with a toolkit of influence-based poisoning attacks. The paper proposes specific data poisoning schemes for algorithms like Logistic regression, random forests, and Naive Bayes.

**Questions:**

Can you clarify (3) and (4) in the weakness.

**Limitations:**

Some of the improvements over prior work are specific to the learning tasks. (For example, the exact choice of influence function for logistic regression.)

The empirical results seem to be specific for delta=0 setting.


**Strengths And Weaknesses:**

Strengths:

1. I found the idea of estimating the probability over general model spaces as opposed to the probability over a single output to be interesting.

2. I also found the usage of local geometry for logistic regression to estimate the poisoning example to be very interesting.

Weaknesses:

1. The framework of auditing seems to be identical to prior work. So, the paper may have oversold the contribution.

2. Some of the improvements over prior work are specific to the learning tasks. (For example, the exact choice of influence function for logistic regression.)

3. I found the result for objective perturbation to be rather surprising. The fact that ML audit matches the theoretical bound is a bit counter intuitive. The reason being that in objective perturbation, typically the epsilons are split to ensure that both the ratio of Jacobians, and the log offs over the model outputs do not change by much. Unless the split is tuned for the best poisoning attack, I cannot intuitively see the curves to match.

4. Are the results for objective perturbation specific to pure \epsilon DP? If not, then one has to use the bounds from Keifer et al. for the Gaussian distribution.

---

> ### Author Response · Authors · 2022-07-29
> **7BRu Response**
>
> 1. We will make more clear the connection to previous work. We also wish to emphasize that in terms of strict auditing estimation itself, principled improvements are relatively small because they must maintain statistical accuracy of a standard procedure. In particular, each of our prior works mentioned (Bischel et al, Jagielski et al, Nasr et al) are themselves relative improvements upon [Ding et al. Detecting Violations of Differential Privacy]. However, our approach combines important contributions leveraged from multiple prior works as well as our own improvements. Prior work we are aware of for auditing a ML model are also highly specialized to one specific model, e.g., for DP-SGD based mechanisms, whereas we are the first to design a procedure to audit general ML models. See response to K5aD for additional detail.
>
> 2. Yes, this is correct - best results are obtained by selecting the appropriate bound $\mathcal{C}$ and summary $\tau$ as a part of our framework.
>
> 3.  We agree with your astute comment that for objective perturbation the theoretical and auditing curves should have a gap because epsilon is split. This is actually what we do observe (Fig 2, dotted grey and blue curves in the leftmost column). We also clarify that the final point on the curves (true epsilon=50) only match because we hit the detection limit with $N=10000$ samples at around y=8.
>
> 4. Yes our objective perturbation method is pure $\epsilon$ DP. For this work all methods we evaluate use standard implementations with delta=0, with the exception of DP-SGD. Our method can be readily adapted for delta > 0 by keeping the delta in the auditing objective of Section 4. Then the objective splits into two quantities, the first of which is what we currently bound. For the second we can use a binomial proportion CI on the denominator $P(M(D') \in S)$ while setting $\delta$ to be the level we wish to audit at. (If we don't have prior information we could also vary delta which results in a curve of estimated $\hat\varepsilon$ vs $\delta$).
>
> Nonetheless, in practical applications, $\delta$ is $o(1/n)$ yielding values which are exceedingly difficult to measure (and which essentially have no effect on our measurements). For example, the minimum probabilities which we can measure with confidence via Monte Carlo are around $p=0.01$ while $\delta$ could be 0.001 for a 1000-point dataset. This barely moves the scale so to speak. For this reason even if we had used a Gaussian mechanism with small $\delta$ we would not expect significantly different results. We will add this discussion in a supplemental section.

---

> > ### Comment · Reviewer_7BRu · 2022-08-09
> > **Thank you for your reply**
> >
> > Thanks for the clarification to my questions. I am increasing my score.

---

### Author Response · Authors · 2022-08-06
**Check-in**

Dear reviewers, we hope you are all doing well. We hope to gently remind you of our rebuttals, as we are pleased with the improvement of the paper from your valuable feedback. We are pleased all initial reviews agreed on the quality of our work and that it should be accepted, and we hope our answers and revision have strengthened that confidence.

Please let us know if there are any remaining questions regarding our work.

---

### Meta-Review · Area_Chair_8Lma · 2022-08-23

**Recommendation:** Accept
**Confidence:** Certain

**Metareview:**

The paper got overall positive reviews. I recommend the authors to carefully incorporate all the pointes raised during the rebuttal period. especially, in regards to the discussion on novelty.

**Award:**

No

---

### Decision · Program_Chairs · 2022-09-14

Accept